

# Growth-promoting characteristics of potential nitrogen-fixing bacteria in the root of an invasive plant *Ageratina adenophora*

Kai Fang[1,2,3,*], Zhu-Shou-Neng Bao[1,3,*], Lin Chen[1,2,3], Jie Zhou[1,3], Zhi-Ping Yang[1,3], Xing-Fan Dong[1,3] and Han-Bo Zhang[1,2,3]

[1] State Key Laboratory for Conservation and Utilization of Bio-Resources in Yunnan, Yunnan University, Kunming, Yunnan Province, China
[2] School of Ecology and Environmental Science, Yunnan University, Kunming, Yunnan Province, China
[3] School of Life Sciences, Yunnan University, Kunming, Yunnan Province, China
* These authors contributed equally to this work.

Corresponding author
Han-Bo Zhang, zhhb@ynu.edu.cn

## ABSTRACT

Root endophytic nitrogen-fixing bacteria (reNFB) have been proposed as important contributors to the invasiveness of exotic legumes; however, the reNFB of invasive nonlegumes has received less attention. In particular, the growth-promoting effect of reNFB on invasive plants remains unknown. In this study, 131 strains of potential nitrogen-fixing bacteria were isolated and purified from the roots of the invasive plant, *Ageratina adenophora*, in Southwest China. Phylogenetically, these reNFB were categorized into three phyla at 97% sequence identity that included Proteobacteria (92.4%), Actinobacteria (4.6%), and Firmicutes (3.1%). The dominant isolates ranked by number were *Pseudomonas* (80 isolates, 61.1%), *Rhizobium* (12 isolates, 9.2%), and *Duganella* (11 isolates, 8.4%). The community composition and diversity of *A. adenophora* reNFB were markedly different across study regions. The capacity of these reNFB to accumulate indolyl-3-acetic acid (IAA), solubilize phosphate, and produce siderophores was determined. All 131 isolates of reNFB accumulated IAA, 67 isolates solubilized phosphate, and 108 isolates produced siderophores. Among the three dominant genera of reNFB, *Pseudomonas* had the highest phosphorus solubilization and siderophore production, while the accumulation of IAA in the genus *Duganella* was the lowest. Interestingly, the calculated reNFB Shannon diversity index of each *A. adenophora* individual was negatively correlated with the capacity of reNFB to produce growth-promoting products. Six randomly selected isolates from three dominant genera were further used to conduct inoculation experiments, and all isolates showed significant positive growth-promoting effects on *A. adenophora* seedlings. The contribution of reNFB to the root biomass was higher than that to the shoot biomass. Our results suggest that reNFB, similar to soil or nodular nitrogen-fixing bacteria, can potentially promote plant growth and may play an important role in the invasion of nonleguminous plants. More detailed studies on the correlation between reNFB and invasive plants are necessary.

## INTRODUCTION

Ongoing globalization and climate warming have enhanced the migration of species (*Lehan et al., 2013*; *Walther et al., 2002*) and therefore have exacerbated the colonization of invasive plants in new habitats (*Moles, Gruber & Bonser, 2008*). Invasive plants that thrive in new environments have been reported to inhibit local plants from competing for more living resources through allelopathy (*Bais et al., 2003*) and other mechanisms (*Hierro, Maron & Callaway, 2005*). Moreover, the invaded ecosystems tend to have higher net primary production compared to native ecosystems (*Wilsey & Wayne Polley, 2006*), indicating that the growth requirements of invasive plants have a high environmental capacity. Invasive plants promote the development of their own communities through highly efficient nutrient cycling (*Ehrenfeld, 2003*). *Liao et al. (2008)* reviewed that the pools of carbon and nitrogen in the invaded systems were significantly increased. Thus, nutrient improvement can be proposed as an important contributor to the success of invaders over native flora. Previously, increased nitrogen input was well documented to facilitate the invasion of exotic species in new habitats (*Siemann & Rogers, 2003*) and enhance the dominance of invasive plants (*Brooks, 2003*).

Recent studies have indicated that the newly established relationship between hosts and local microbes have greatly impacted the competitiveness of invasive plants (*Van Der Putten, Klironomos & Wardle, 2007*). For example, it has been suggested that invasive plants maintain long-term dominance by accumulating local pathogens (*Eppinga et al., 2006*) or beneficial microbes (*Rout & Callaway, 2009*; *Van Der Putten et al., 2013*). Most invasive plants can form mutualistic relationships with local microbes, of which symbiotic nitrogen-fixing bacteria (NFB) play an extremely important role in gaining additional nutritional resources for invasive plants that are competing with native plants (*Rodríguez-Echeverría & Traveset, 2015*). Currently, the associations between root endophytic nitrogen-fixing bacteria (reNFB) and invasive legumes are widely studied. For example, *Lafay & Burdon (2006)* showed that the invasive leguminous plant *Cytisus scoparius* in Australia was able to form nodules with *Bradyrhizobium*, *Rhizobium*, and *Mesorhizobium*. *Parker, Malek & Parker (2006)* further demonstrated that the biomass of *C. scoparius* when inoculated with *Bradyrhizobium* was twice that of a noninoculated plant. *Wei et al. (2009)* also demonstrated that *Robinia pseudoacacia*, a leguminous plant from North America, could invade China's barren regions by forming nodules with bacteria in local soils.

However, in addition to the root nodules of legumes, a variety of nitrogen-fixing bacteria have been isolated in the roots, stems, and leaves of nonlegumes, which are named endophytic diazotrophs (*Olivares et al., 1996*). These endophytic diazotrophs have been reported as an important mechanism in facilitating the invasion of the exotic plant *Sorghum halepense* in grasslands where nitrogen sources are extremely scarce (*Rout & Chrzanowski, 2009*). In addition to nitrogen fixation, endophytic diazotrophs have also been shown to promote plant growth with their secondary metabolites (*Carvalho et al., 2016*), including indolyl-3-acetic acid (IAA) accumulation, phosphorus solubilization, and siderophore production (*Chauhan, Bagyaraj & Sharma, 2013*; *Rout et al., 2013*). Therefore, studies on the relationship between endophytic diazotrophs and invasive plants are important for better

understanding the mechanisms underlying the colonization and dominance of invasive plants in nonnative habitats. Unfortunately, thus far, little is known about endophytic diazotroph communities of invasive nonlegumes in a relatively large introduced geographic range or their effects on promoting the growth of invasive plant seedlings.

*Ageratina adenophora* (Sprengel) R. M. King and H. Robinson is a perennial herb of the Compositae family that is native to Central America but a noxious weed in Asia, Africa, Oceania, and Hawaii. It first invaded Yunnan Province, China, in the 1940s and is now widely distributed in Yunnan, Guizhou, Sichuan, Guangxi, and Tibet provinces in Southwest China and has continuously spread east- and northward at a rate of approximately 20 km/year (*Wang & Wang, 2006*). *A. adenophora* has shown a strong acclimation capacity to a wide range of nitrogen environments (*Wang & Feng, 2005*), and its competitive ability could be promoted by artificially applying nitrogen (*Zhao, Meng & Li, 2007*). Furthermore, *Niu et al. (2007)* showed that available soil nutrients (N, P, and K) were greatly increased in soils that were heavily invaded by *A. adenophora*. *Xu et al. (2012)* demonstrated that, compared to noninvaded habitats, *A. adenophora* accumulated more abundant and diverse nitrogen-fixing bacteria inhabiting rhizosphere soils in heavily invaded areas, suggesting a potential role of the nitrogen cycle, which is driven by bacteria, in the invasiveness of *A. adenophora*. However, no study has been conducted on the investigation of endophytic diazotrophs of *A. adenophora*, as well as their effects on promoting the growth performance of seedlings.

In this study, we first isolated endophytic diazotrophs of *A. adenophora* roots, named potential reNFB, in a large geographic range and attempted to explore the community composition and phylogeny of *A. adenophora* reNFB from different regions. Second, we detected the growth-promoting products of all obtained reNFB, including IAA accumulation, phosphorus solubilization, and siderophore production, and analyzed the differences in the ability of dominant reNFB to produce growth-promoting products. Finally, we carried out an inoculation experiment with the focal isolates to verify the growth-promoting effects of reNFB on the seedlings of *A. adenophora*.

# MATERIALS AND METHODS

## Isolation and identification of reNFB

### Root collection

We collected *A. adenophora* roots from five regions in Yunnan Province (22°37′–25°52′N, 99°18′–104°10′E, 1,708–2,128 m), Southwest China, and named them XM, CY, XS, WS, and YL (Table 1). All of the sampling sites were along roadsides and had been badly invaded by *A. adenophora* with coverage of more than 80% (visually estimated). In August 2016, three mature and healthy *A. adenophora* were selected in each region, and the whole plant was dug up with a shovel and then brought back to the laboratory together with its rhizosphere soils. The roots were used to isolate endophytic bacteria within 3 days after collection.

### Bacterial isolation and molecular identification

We cut 10 randomly selected fine-root fragments from each harvested *A. adenophora* individual, rinsed them with tap water in empty petri plates and surface sterilized them as

Table 1 Geographical description and vegetation of study regions.

| Region[a] | Altitude (m) | Latitude | Longitude | Dominant plants |
|-----------|--------------|----------|-----------|-----------------|
| XM | 1,715 | 99.80 | 22.62 | *Sida acuta, Urena lobata, Rubus corchorifolius* |
| CY | 1,708 | 99.30 | 23.22 | *Rubus corchorifolius, Betula alnoides, Pistacia weinmannifolia, Desmodium sequax* |
| XS | 2,040 | 102.37 | 24.50 | *Pinus yunnanensis Franch, Alnus nepalensis, Myrica rubra, Amphicarpaea edgeworthii* |
| WS | 1,980 | 104.17 | 24.52 | *Desmodium sequax* |
| YL | 2,128 | 99.43 | 25.87 | *Pinus yunnanensis Franch, Desmodium heterocarpon* |

**Note:**
[a] XM, Ximeng county; CY, Cangyuan county; XS located at Kunming city; WS, Wenshan Zhuang and Miao Autonomous Prefecture; YL, Yunlong county.

follows: fine roots were submerged in 95% ethanol for 10 s and 5% sodium hypochlorite for 3 min and were rinsed with sterile water three to five times, and then the surface water of the roots was dried with sterile filter paper. Roots were printed on nutrient agar culture medium (1,000 ml distilled water containing beef extract one g, peptone five g, yeast extract two g, NaCl five g, and agar 15 g, pH 6.8–7.0) to confirm that the surface sterilization was successful. Then, the disinfected fine roots were cut into fragments of approximately 0.5 cm for detection of endophytic bacteria. In total, 150 root fragments were subjected to testing. The thoroughly surface-sterilized root fragments were placed in separate centrifuge tubes containing 150 µl of sterile water, crushed with a sterilized pipette tip, and streaked with inoculation tools onto modified yeast-mannitol agar (YMA) culture medium (1,000 ml distilled water containing D-mannitol 10.00 g, yeast extract 3.00 g, $MgSO_4 \bullet 7H_2O$ 0.20 g, NaCl 0.10 g, $K_2HPO_4$ 0.25 g, $KH_2PO_4$ 0.25 g, agar 15.00–18.00 g, and 4% Congo red one ml, pH 6.8–7.0) (*Kanu & Dakora, 2012*). The plates were sealed with parafilm and then cultured for 3–5 days at 28 °C to form colonies. The assay of each root fragment was repeated three times, and a total of 450 petri dishes were used to isolate endophytic bacteria of *A. adenophora* in this experiment. After 3–5 days of culture, a single colony with good growth and without Congo red absorption was selected for purification.

Numerous studies have shown that bacteria that can grow normally in medium lacking nitrogen have the potential to fix nitrogen (*Ding et al., 2005*; *Park et al., 2005*; *Silva et al., 2011*). In order to screen out the candidate bacteria with nitrogen fixation from the endogenous bacteria obtained by the above steps, the purified isolates were inoculated into nitrogen-free liquid medium (1,000 ml distilled water containing $KH_2PO_4$ 0.2 g, $MgSO_4 \bullet 7H_2O$ 0.2 g, NaCl 0.2 g, $CaCO_3$ 5.0 g, $C_6H_{14}O_6$ 10.0 g, and $CaSO_4 \bullet 2H_2O$ 0.1 g, pH 7.0) (*Ashby, 1907*) in a glass test tube covered with a plastic stopper to confirm their potential for nitrogen fixation, and the isolates capable of being continuously cultured for seven generations with 24 h per generation were preliminarily identified as nitrogen-fixing bacteria (Dataset S1) (*Aeron et al., 2015*).

The reNFB of *A. adenophora* selected from above were inoculated into test tubes containing five ml yeast-mannitol (YM) liquid culture medium and cultured for 24 h under the conditions of 28 °C and 180 r/min. Then, one ml of bacterial suspension was taken and centrifuged for 1 min at 12,000 rpm to obtain the bacterial precipitate. Total genomic DNA was extracted directly from the bacterial precipitate of each isolate using an AxyPrep Bacterial Genomic DNA Miniprep Kit. The primers 27F (5′-CGA AGT AGT

TTG ATC CTG GCT-3′) and 1523R (5′-AGG AGG TGA TCC AGC CGC A-3′) were used to amplify approximately 1,500 bp of 16S rRNA (*Edwards et al., 1989*). Each 50 μl PCR included five μl 10× amplification buffer, five μl dNTP mixture (two mM), 0.25 μl Taq DNA polymerase (two U/μl), one μl each primer (10 μM), one μl template DNA, and 37.75 μl ddH$_2$O. The amplification was run in a Veriti® 96 Well Thermal Cycler (Applied Biosystems Inc., Foster City, CA, USA) (3 min at 94 °C, followed by 35 cycles of 30 s at 94 °C, 45 s at 54 °C, and 100 s at 72 °C, and 7 min at 72 °C). PCR products were tested by 1% gel electrophoresis and then sent to the Beijing Genomics Institute for bidirectional sequencing.

We processed the raw sequence data with SeqMan software (DNAStar 7.1, Madison, WI, USA) to pair the merged bidirectional sequencing DNA, resulting in a single sequence with a length of approximately 1,400 bp. Based on the GenBank database, we carefully conducted sequence homology analysis and sequence correction and then cut out chimeric bases using ClustalX 2.1 (*Thompson et al., 1997*). After that, we grouped the resulting sequences that were approximately 1,300 bp long into operational taxonomic units (OTUs) at a 97% sequence identity with Mothur 1.35.1 (Dataset S1) (*Schloss et al., 2009*).

### Phylogenetic analysis

We randomly selected representative isolates from each OTU, and the closest relatives of the representative isolates were searched and downloaded in the GenBank nr/nt database (2018.06) (Dataset S1). Phylogenetic trees were constructed with the sequences of both representative isolates and reference strains using MrBayes 3.2.5 (*Huelsenbeck & Ronquist, 2001*) and raxmlGUI 1.5b1 (*Silvestro & Michalak, 2012*). Bayesian analysis was performed in two million operations with the model GTR+T+G and parameters nst = 6 and rates = gamma, which was calculated by the jModelTest 2.0 (*Darriba et al., 2012*; *Guindon & Gascuel, 2003*), and maximum likelihood was carried out in 1,000 replicates through rapid bootstrap analysis. Phylogenetic trees were edited in FigTree v1.4.3 (*Rambaut, 2016*) and displayed by combining them into one figure with Adobe Illustrator CC 2015 (Adobe Systems Inc., San Jose, CA, USA). The nucleotide sequences reported in this study have been deposited in GenBank under accession numbers MK249666–MK249688 and MK574703–MK574810.

## Growth-promoting products of reNFB

### IAA accumulation test

Salkowski's reaction was used to test IAA accumulation of *A. adenophora* reNFB at 72 and 144 h of culture, respectively (*Loper & Schroth, 1986*). The suspension was prepared by culturing bacteria in YM liquid medium at 28 °C and 180 r/min for 24 h. Then, five μl of bacterial suspension was inoculated into five ml of liquid King's B medium (five ml glycerine containing peptone 20.0 g, K$_2$HPO$_4$ 1.5 g, and MgSO$_4$•7H$_2$O 1.5 g, pH 7.2 ± 0.2) containing L-TRP (0.5 g/ml), which were incubated at 28 °C and 180 r/min for 72 and 144 h, respectively. After that, bacteria cells were removed by centrifugation for 5 min at 12,000 rpm, and 500 μl supernatant and 500 μl Salkowski reagent (ferric chloride 12 g, concentrated sulfuric acid 430 ml, and deionized water 570 ml) were mixed and allowed to

react for 30 min in the dark prior to the absorbance value being measured at 530 nm in a SpectraMax® 340PC$^{384}$ (Molecular Devices Inc., San Jose, CA, USA). King's B medium that was not inoculated was used as a reference instead of supernatant. The assay was repeated three times for each isolate. After subtracting the reference absorbance values from the sample, the IAA content of each isolate was calculated using the regression curve of the IAA standard (Dataset S1).

### Phosphate solubility test

Phosphate solubility was determined by measuring the diameters of the transparent circle formed by dissolution of phosphate on glucose yeast (GY) medium (1,000 ml distilled water containing glucose 10 g, yeast powder two g, and agar 15 g, pH 7.0) containing $K_2HPO_4$ (50 ml distilled water containing $K_2HPO_4$ 5 g) and $CaCl_2$ (100 ml distilled water containing $CaCl_2$ 10 g) (Sylvester-Bradley et al., 1982). Five microliters of bacterial suspension, which was activated by YM liquid medium, was inoculated into five ml of liquid King's B culture medium for 48 h at 28 °C and 180 r/min, and then five μl of King's B culture with bacteria was inoculated onto the GY test plate. Sterile water was used as a control instead of a bacterial suspension. The assay was repeated four times for each isolate. The ratio of the transparent circle diameter to the colony diameter indicated the phosphate solubility index of each isolate, and solubility indices equal to 1 were considered negative (Dataset S1).

### Siderophore production test

A chrome azurol S (CAS) liquid assay was used to test siderophore production (Machuca & Milagres, 2003). The 100 μl CAS assay solution, prepared according to Schwyn & Neilands (1987), was injected into the 100 μl centrifugal supernatant of King's B bacterial culture (as mentioned in the IAA accumulation test), which reacted for 60 min at room temperature, and then the absorbance value was measured at 630 nm in a SpectraMax® 340PC$^{384}$ (Molecular Devices Inc., San Jose, CA, USA). Liquid King's B medium was used as a reference instead of supernatant. The assay was repeated three times for each isolate. Siderophore units were calculated by $(Ar - As)/Ar$, where $Ar$ and $As$ represent the absorbance values of the reference and sample, respectively, and percentages of siderophore units less than 10% were considered negative (Machuca & Milagres, 2003) (Dataset S1).

## Growth-promoting effects of reNFB

According to phylogenetic analysis, six isolates were randomly selected from three dominant genera to study their effects on the growth of *A. adenophora* seedlings, including two *Pseudomonas* (WS29 and WS14), three *Rhizobium* (XM06, XM04, and WS05) and one *Duganella* (XS01). Seeds and soil were, respectively, collected from wild *A. adenophora* and cropland in Kunming city, Yunnan Province, Southwest China. Seeds were surface sterilized by submerging them in 4% sodium hypochlorite for 5 min and were rinsed with sterile water five to six times. A substrate that included cropland soils, humus, and vermiculite were mixed at a proportion of 3:2:1 and sterilized three times at 24 h intervals by autoclaving (121 °C, 0.135 MPa, 2 h). Five milliliters of bacterial suspension ($10^9$ CFU/ml) of each isolate mixed with 50 ml of sterile water was separately injected into

a sterilized plastic cup (700 ml) containing 20 g sterilized soil mixture and sown with approximately 30 surface-sterilized seeds of *A. adenophora* in each cup, and the cup was sealed with sterilized PTFE bacteria filter membranes. The assay was repeated three times for each isolate, and the control group was not inoculated with bacterial suspension. Seeds were germinated in an RXZ-380D growth chamber (Ningbo Southeast instrument Co., Ltd, Ningbo, China) with a temperature of 25/20 °C (day/night), light intensity of 12,000 lux for a 12-h photoperiod and humidity of 60%.

After 40 days, seedlings were thinned out of each cup, and only one seedling with the best growth was kept, and the remaining seedlings were harvested on the 60th day of continuous growth. The surface soil of harvested seedlings was removed with an SB3200 ultrasonic instrument (Branson Ultrasonics Co., Ltd, Shanghai, China), and the surface water was dried with filter paper. After that, the length and wet biomass of shoots and roots were measured, and their dry biomass was measured after drying at 60 °C for 48 h (Dataset S1).

## Statistical analyses

R 3.5.2 was used for random sampling among all isolates, and the OTU rarefaction curve at 97% sequence identity was plotted in Excel 2013 (Microsoft Inc., Redmond, WA, USA). The Shannon diversity index of each *A. adenophora* individual was calculated using Excel 2013 (Microsoft Inc., Redmond, WA, USA) as previously described (*Hill et al., 2003*). One-way ANOVA was used to compare Shannon diversity across different study regions. A nonparametric test was used to compare the production capacity of growth-promoting products between sampling areas. To meet the normality assumption, log transformations were applied to IAA accumulation and the phosphorus solubility index, and a square-root transformation was applied to siderophore units. One-way ANOVA was used to calculate these three kinds of growth-promoting products produced by *A. adenophora* reNFB across the dominant genus, and the homogeneity test of variance passed or failed to select Duncan or Dunnett T3 for pairwise comparison. In addition, we performed linear regression analysis between the Shannon diversity index of each *A. adenophora* and their reNFB growth-promoting product productivity.

Log transformations were applied to the root and shoot biomass of the experimental seedlings to meet normality and homoscedasticity assumptions. Independent-samples *t*-test was used to compare the different effects of each bacterial treatment and sterilized control on the growth performance of *A. adenophora* seedlings. In addition, we calculated the response index (RI) to evaluate the effect of reNFB on the shoots and roots of *A. adenophora* seedlings. RI was calculated as follows: $(\text{variable}_{\text{inoculation}} - \text{variable}_{\text{sterility}})/\text{variable}_{\text{sterility}}$ (*Williamson & Richardson, 1988*), where $\text{variable}_{\text{inoculation}}$ and $\text{variable}_{\text{sterility}}$ were the shoot or root length and biomass of each cup (replicates), with inoculation treatment and sterilized control, respectively. Normality assumptions were not met; thus, nonparametric tests were used to compare the differences in RI between shoots and roots.

In addition to the calculation of Shannon diversity, all other statistical analyses were conducted by using SPSS 22.0 (SPSS Inc., Chicago, IL, USA). In addition to the OTU

rarefaction curve, all other figures were created using GraphPad Prism 7 (GraphPad Software Inc., San Diego, CA, USA).

## RESULTS

### Species composition of reNFB

In total, 150 isolates were obtained from the roots of 15 *A. adenophora* individuals across the five regions mentioned above by using YMA selective medium, but only 131 isolates were able to grow continuously for seven culture generations in nitrogen-free medium, including 29 isolates from XM, 24 from CY, 23 from XS, 27 from WS, and 28 from YL. In the phylogenetic tree (Fig. 1), these isolates were placed into three phyla at 97% sequence identity and included Proteobacteria (92.4%), Actinobacteria (4.6%), and Firmicutes (3.1%). All isolates in Proteobacteria were further divided into three subgroups, including alpha (18.2%), beta (14.1%), and gamma (67.8%). At the genus level, the dominant isolates were *Pseudomonas* (80 isolates, 61.1%), followed by *Rhizobium* (12 isolates, 9.2%) and *Duganella* (11 isolates, 8.4%). Other genera had an occurrence frequency of less than 3.0%.

At the genus level, there were clear differences in the composition of *A. adenophora* reNFB among the study regions (Fig. 2A). WS contained the most abundant isolates (11 genera), while YL had the least abundant isolates (three genera).With the exception of XS, in which *Duganella* (39.1%) was the most abundant reNFB, *Pseudomonas* was the most dominant reNFB in the other four regions, accounting for 65.5% in XM, 83.3% in CY, 44.4% in WS, and 89.3% in YL. The Shannon diversity index calculated at 97% of the identity level showed that the diversity of XS and WS were significantly higher than that of CY, XM, and YL (Fig. 2B).

### Growth-promoting products of reNFB

Of the 131 reNFB isolated in this experiment, 66 isolates were able to simultaneously accumulate IAA, dissolve phosphate and produce siderophores, including 57 *Pseudomonas* sp., three *Rhizobium* sp., one *Duganella* sp., one *Agrobacterium* sp., one *Nocardioides* sp., one *Variovorax* sp., one *Pseudarthrobacter* sp., and one *Paenibacillus* sp. All 131 reNFB could accumulate IAA, and the content of IAA in most isolates gradually increased with increasing incubation time. For example, the number of isolates with a yield of IAA greater than 10 μg/ml increased from 19 isolates with an incubation time of 72 h to 29 isolates with an incubation time of 144 h (Figs. 3A and 3B). A total of 67 isolates of reNFB were able to dissolve phosphate, accounting for 51.15% of the total isolates, mainly distributed in XM, CY, and YL (Fig. 3C). There were 108 isolates that could produce siderophores (siderophore units greater than 10%), but only 10 isolates in XS could produce siderophores (Fig. 3D). In addition, the ability of *Pseudomonas* to dissolve phosphate and to produce siderophores was relatively high among the three dominant genera (Figs. 3G and 3H), while the accumulation of IAA in the genus *Duganella* was the lowest among the three dominant genera (Figs. 3E and 3F). Interestingly, the calculated reNFB Shannon diversity index of each *A. adenophora* individual was negatively correlated with the capacity of reNFB to produce growth-promoting products, and all were statistically significant, with the exception of IAA at 144 h (Fig. 4).

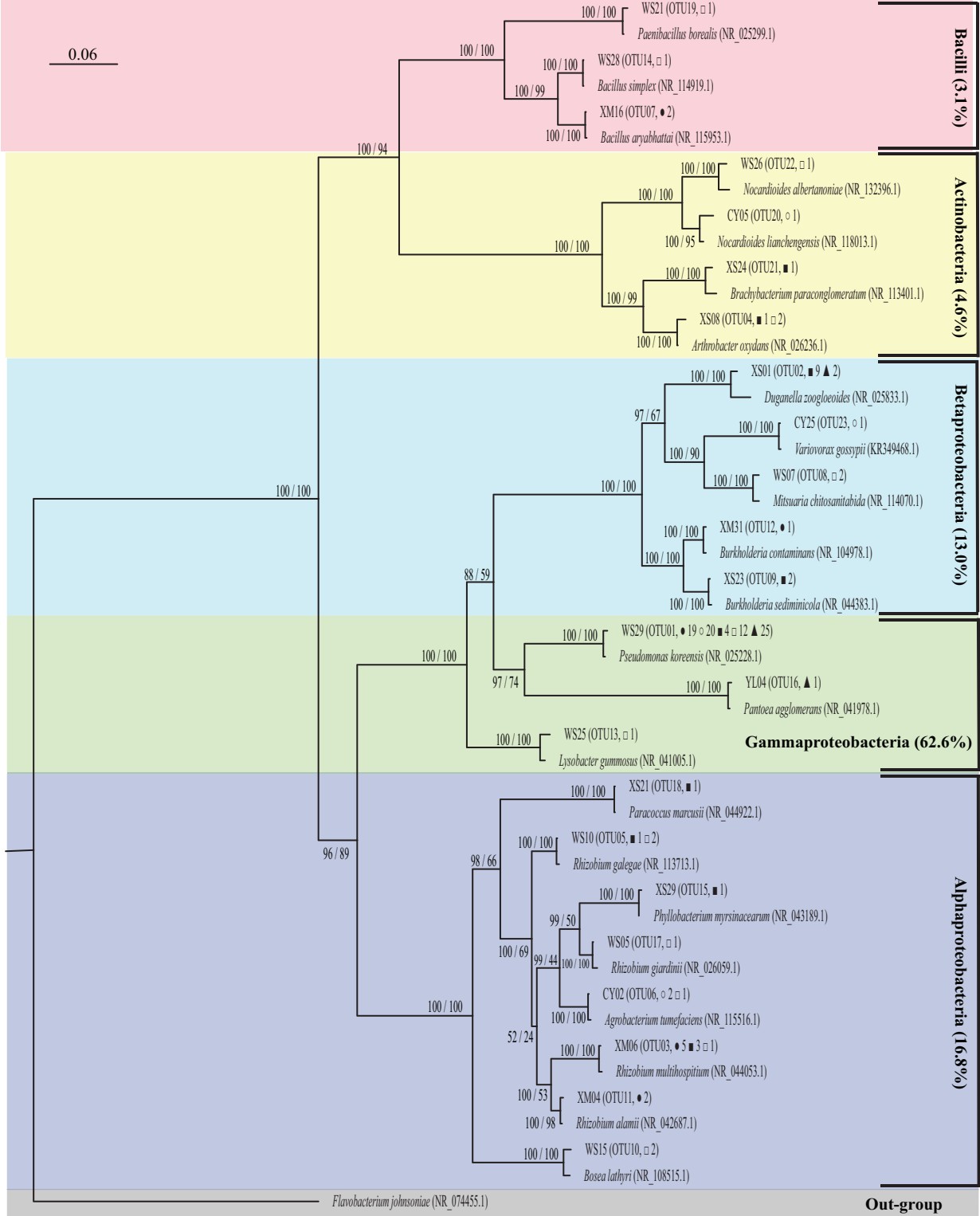

**Figure 1 Phylogenetic tree of representative potential reNFB OTU of *A. adenophora* and their best blast matches showing their phylogenetic affinities.** The bootstrap values and posterior probability are indicated at the branch node, shown as MrBayes/maximum likelihood. Each OTU and its occurrence time in XM (solid circle), CY (open circle), XS (solid square), WS (open square), and YL (solid triangle) regions is indicated in parentheses, as well as the GenBank accession numbers of reference sequences. The scale bar represents 6% estimated sequence divergence. *Flavobacterium johnsoniae* (NR_074455.1) was used as the out-group.

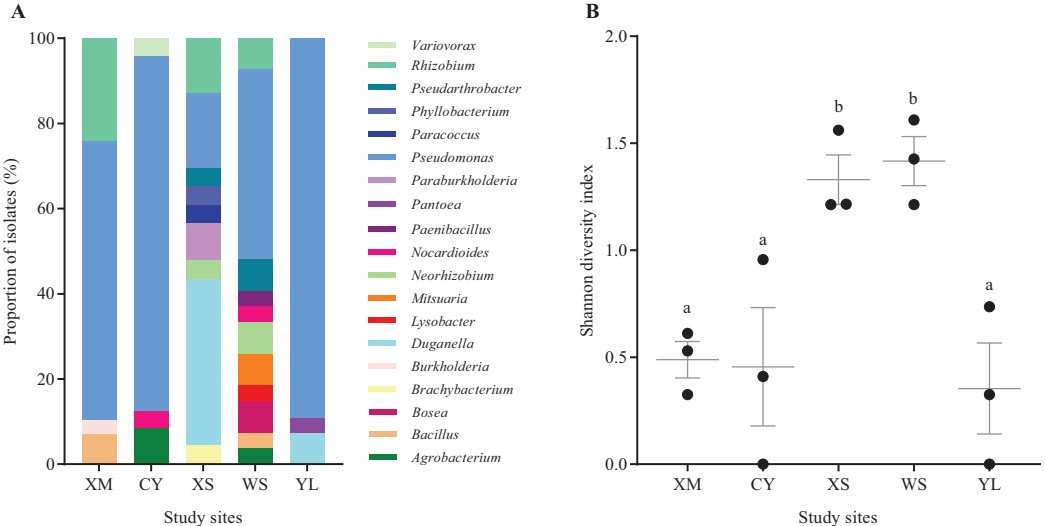

**Figure 2** **The species composition at the genus level (A) and the Shannon diversity index at the 97% identity level (B) of *A. adenophora* reNFB from different study regions.** Error bars depict the standard error. Black dots show the values of the Shannon diversity index. One-way ANOVA was used to compare Shannon diversity indices across different study regions ($F = 8.628$, $P = 0.003$), and the homogeneity test of variance was passed ($P = 0.412$), and Duncan's multiple range test was used for pairwise comparison. Different lowercase letters indicate that the difference was significant, and identical lowercase letters indicate nonsignificant differences.

## Effects of reNFB on the growth of *A. adenophora* seedlings

Root endophytic nitrogen-fixing bacteria differed in their effects on the growth performance of *A. adenophora* seedlings (Fig. 5). The average shoot length of *A. adenophora* seedlings treated with *Duganella* sp. (OTU02; XS01) was lower than that of the control group, all others revealed positive effects on various plant growth metrics. The WS05 isolate (*Rhizobium* sp.) was the best growth-promoting isolate of *A. adenophora* among these six reNFB, which greatly increased the length of shoots ($P < 0.001$) and roots ($P < 0.001$) and had an evident positive effect on shoot biomass ($P_{wet} < 0.001$; $P_{dry} < 0.001$) and root biomass ($P_{wet} = 0.004$; $P_{dry} = 0.005$). Compared to the shoots, the six selected isolates showed a stronger positive effect on the roots of seedlings, and the wet- and dry-biomass RI of roots were significantly higher than that of shoots (Fig. 6).

## DISCUSSION

Exotic plants are able to change soil nitrogen cycling in an invaded range (*Rout & Callaway, 2009*), and the increase in available soil nitrogen can improve the competitiveness of invasive plants (*Liao et al., 2008*). These facts undoubtedly imply that the abundance and diversity of nitrogen-fixing bacteria may be important factors for plant invasion (*Rodríguez-Echeverría, Crisóstomo & Freitas, 2007*). Until now, many efforts have focused on nitrogen-fixing bacteria of invasive legumes (*Lafay & Burdon, 2006*; *Parker, Malek & Parker, 2006*); however, the characterization of nitrogen-fixing bacteria in invasive non-legumes has been limited. In this study, 150 endophytic bacteria were initially isolated from roots of invasive *A. adenophora* by using modified YMA culture medium, and 131 isolates were

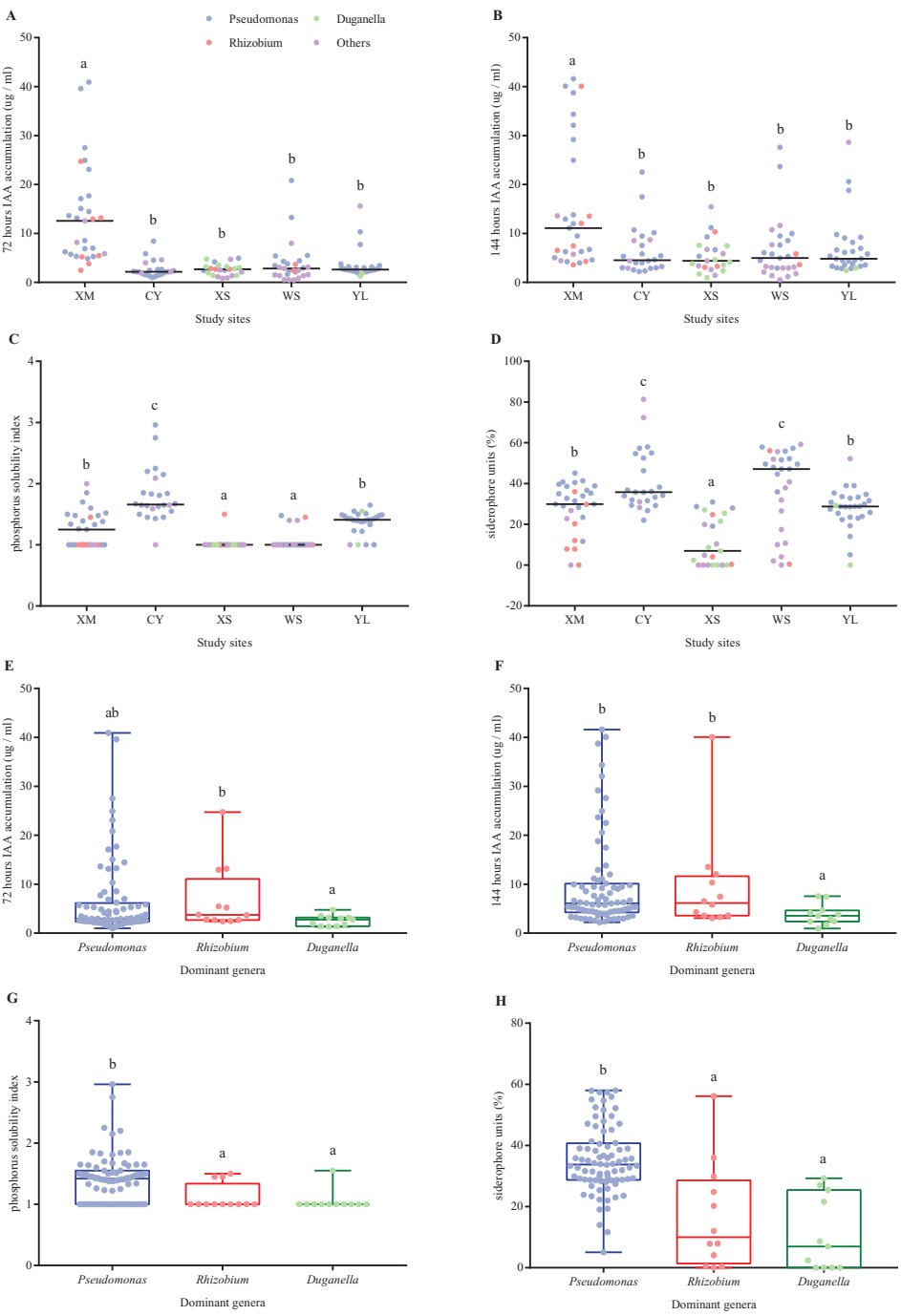

**Figure 3 IAA accumulation (A, B), phosphorus solubility index (C) and siderophore units (D) of the total reNFB from each study region. Differences in IAA accumulation (E, F), soluble phosphorus index (G) and siderophore units (H) among the three dominant genera.** The colorful points represent the values of growth-promoting products of each isolate. The horizontal lines in (A–D) indicate the median number of each study region. The horizontal lines and whiskers in (E–H) represent the mean value and min to max values, respectively. A nonparametric test was used to compare the production capacity of growth-promoting products between sampling areas. Log-transformations were applied to IAA accumulation and the phosphorus solubility index in (E–G), while square-root transformations were applied to siderophore units in (H). Bars with different lowercase letters in each figure are significantly different at $P < 0.05$.                                           

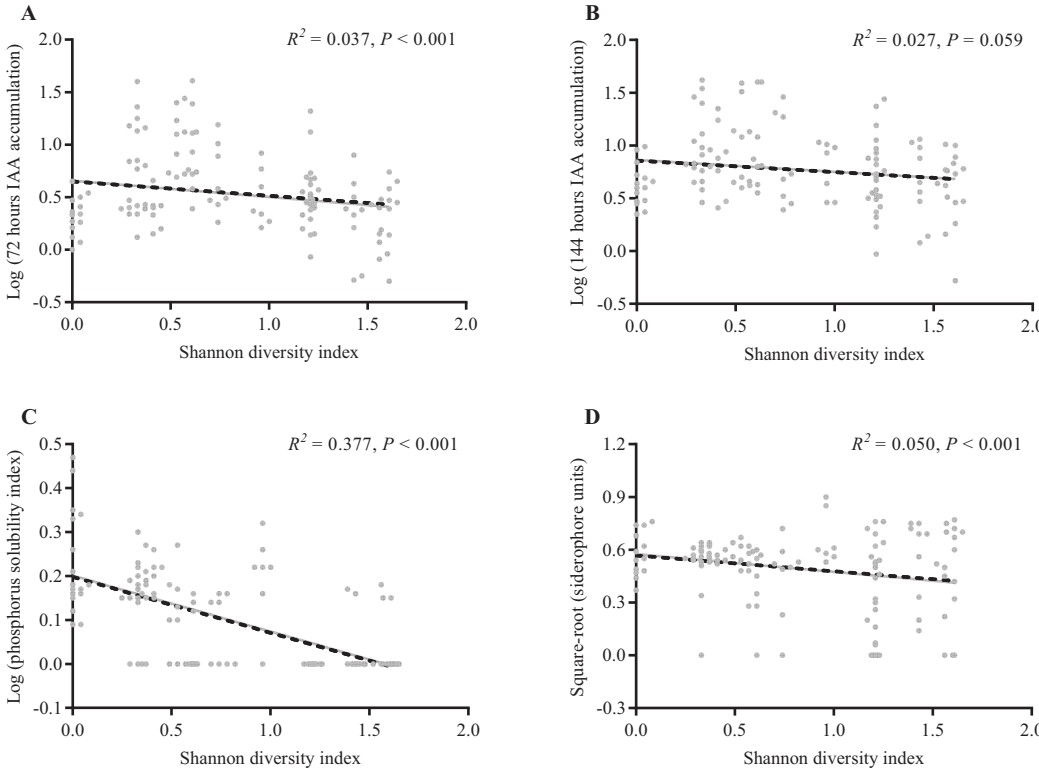

**Figure 4  Linear regression analysis between the Shannon diversity index of each *A. adenophora* and their reNFB growth-promoting product productivity.** Log transformations were applied to IAA accumulation and the phosphorus solubility index in (A–C). Square-root transformations were applied to siderophore units in (D). The equation for the regression of 72 h IAA accumulation and Shannon diversity index is $Y = -0.1367 * X + 0.6502$ (A). The equation for the regression of 144 h IAA accumulation and Shannon diversity index is $Y = -0.1085 * X + 0.8565$ (B). The equation for the regression of phosphorus solubility index and Shannon diversity index is $Y = -0.127 * X + 0.1979$ (C). The equation for the regression of siderophores units and Shannon diversity index is $Y = -0.08961 * X + 0.5673$ (D).

preliminary determined as reNFB by continuous culture for seven generations in nitrogen-free culture medium (*Aeron et al., 2015*). The dominant reNFB of *A. adenophora* was *Pseudomonas* (61.1%), *Rhizobium* (9.2%), and *Duganella* (8.4%) (Fig. 1). *Pseudomonas* and *Rhizobium* have been widely reported as endogenous NFB of nonlegumes (*Carvalho et al., 2016*; *Santi, Bogusz & Franche, 2013*). For example, nitrogen-fixing *Pseudomonas* sp. have been isolated from stem and leaf tissues of sugarcane (*Magnani et al., 2010*), barley seeds (*Zawoznik et al., 2014*), and wild rice roots (*Chaudhary et al., 2012*). In particular, *Pseudomonas* accounted for 61.1% of the reNFB of *A. adenophora*, suggesting a possible important contribution of *Pseudomonas* to *A. adenophora* competitiveness. *Rhizobium* are mainly reported as NFB of legumes (*Lafay & Burdon, 2006*), in nonlegumes, for example, *Biswas, Ladha & Dazzo (2000)* demonstrated that *Rhizobium* could infect rice and increase rice yield. The genus *Duganella* was first proposed by *Hiraishi, Shin & Sugiyama (1997)* as a reclassification of a misnamed strain, and until now, its interaction with plants has not been clear. *Li et al. (2004)* isolated *Duganella violaceinigra* sp. from a forest soil sample collected from Yunnan Province, Southwest China. In this study, *Duganella* were isolated as

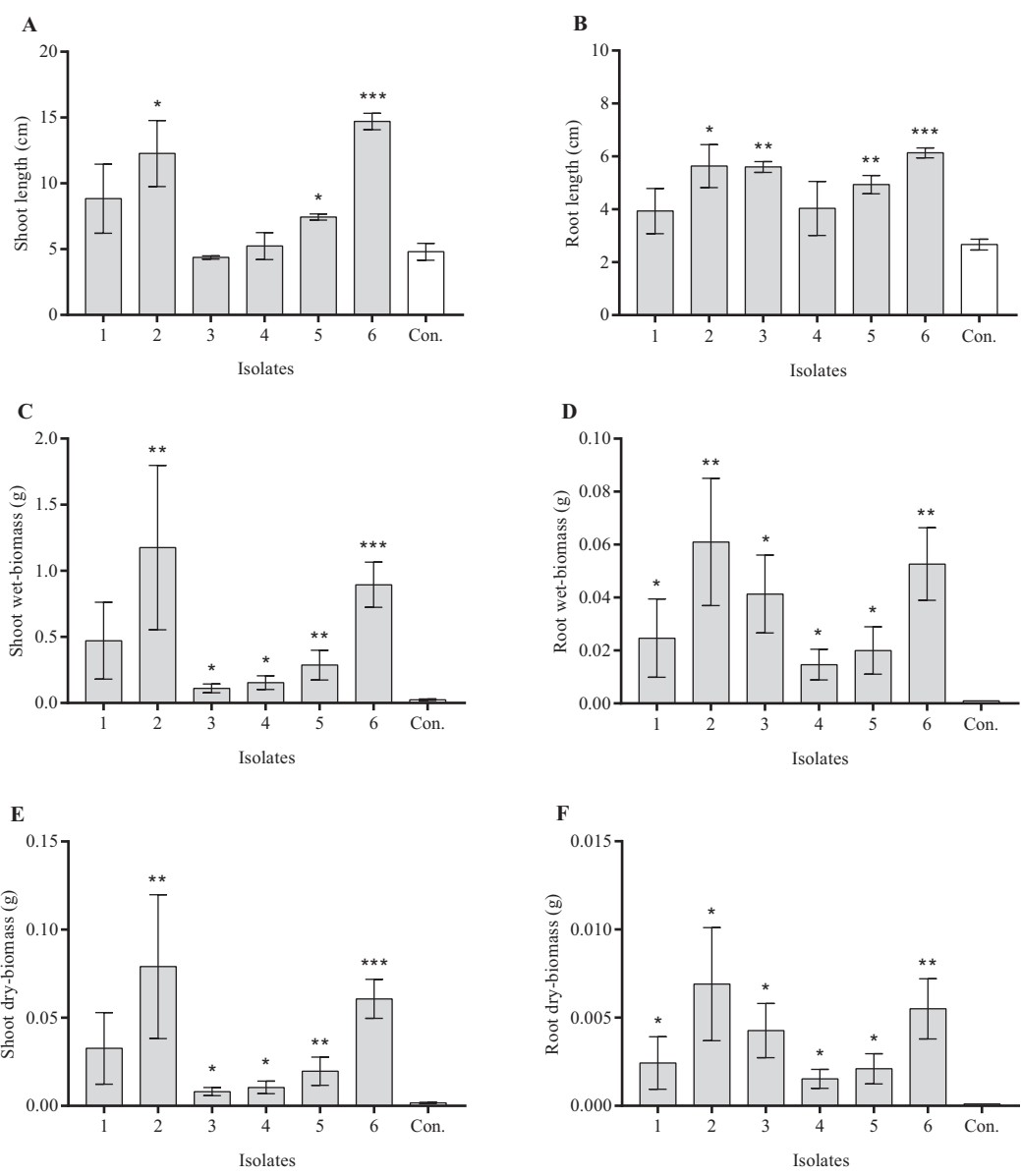

**Figure 5 Effects of focal reNFB on the growth performance of *A. adenophora* seedlings, including the shoot lengths (A), root lengths (B), shoot wet-biomass (C), root wet-biomass (D), shoot dry-biomass (E), root dry-biomass (F).** 1 represents *Pseudomonas* sp. (OTU01; WS29); 2 represents *Pseudomonas* sp. (OUT 01; WS14); 3 represents *Duganella* sp. (OTU02; XS01); 4 represents *Rhizobium* sp. (OTU03; XM06); 5 represents *Rhizobium* sp. (OUT11; XM04); 6 represents *Rhizobium* sp. (OTU17; WS05); and Con. represents sterile water control. Log transformations were applied to wet and dry biomass (C–F). Significant difference is the comparison between treatment and control by using Independent-Samples *t*-test and marked as following (*<0.050, **<0.010, ***<0.001). Error bars depict the standard error for the average growth performances of *A. adenophora*.

reNFB, indicating that soil-dwelling *Duganella* can infect *A. adenophora* and may make an important contribution to *A. adenophora* invasiveness. In addition, compared to another study of nonsymbiotic NFB inhabiting rhizosphere soils of *A. adenophora* in Yunnan Province (*Xu et al., 2012*), *Pseudomonas* and *Rhizobium* accounted for only 8.2% and 7.0%,

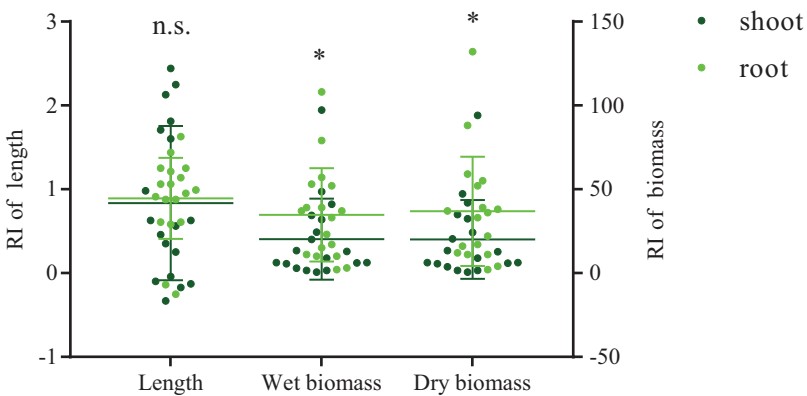

**Figure 6 Response indices (RI) of shoot and root lengths of wet and dry biomass of *A. adenophora* seedlings.** Significant difference is the comparison between shoot and root by using nonparametric tests ($P_{length}$ = 0.521, $P_{wet-biomass}$ = 0.037, $P_{dry-biomass}$ = 0.037) and mark as following (n.s. >0.05, *<0.05). The long horizontal lines represent mean value. Error bars depict the standard deviation for the average RI of *A. adenophora* shoot and root, respectively.

it can be concluded that there are differences between reNFB and nonsymbiotic NFB of *A. adenophora*. Therefore, there might be host specialization in the interaction between *A. adenophora* and reNFB. Although plants can dramatically modify their soil environment to shape the soil microbial communities through the rhizosphere effect (*Bremer et al., 2009*; *Jones, Nguyen & Finlay, 2009*) and improve themselves growth performance, the selective accumulation of specific reNFB, for example, *Pseudomonas*, may be an effective strategy for the growth advantage of *A. adenophora*.

It was previously reported that the endogenous bacteria of the same plant species revealed regional endemism (*Li et al., 2012*), and we also found that there were marked differences in the community composition and diversity of *A. adenophora* reNFB in different regions (Fig. 2). However, the regions in this paper were distributed in a relatively large geographic range, and there were certain differences in environmental background, which might explain the differences in reNFB of *A. adenophora*. However, differences in reNFB diversity might be related to plant growth stage. It was reported that the diversity of functional endogenous bacteria was distinct in different stages of plant growth (*Loaces, Ferrando & Scavino, 2011*). The five regions were distributed in three latitudes, where *A. adenophora* might be in different development stages under certain light and temperature conditions when we sampled in August.

Previous studies have shown that plant endogenous bacteria are able to not only fix nitrogen but also accumulate IAA (*Chauhan, Bagyaraj & Sharma, 2013*), dissolve phosphate (*Kuklinsky-Sobral et al., 2004*) and produce siderophores (*Loaces, Ferrando & Scavino, 2011*). We measured 131 isolates of *A. adenophora* reNFB and found that all isolates accumulated IAA (Figs. 3A and 3B), 67 isolates solubilized phosphorous (Fig. 3C) and 108 isolates produced siderophores (Fig. 3D). Among the three dominant genera, *Pseudomonas* showed the highest growth-promoting product productivity (Figs. 3E–3H). Previously, *Pseudomonas* were reported as excellent producers of various crop growth regulators that stimulate plant growth (*Pastor et al., 2014*) and inhibit pathogen infection

(*Khare & Arora, 2010*). However, *Chauhan, Bagyaraj & Sharma (2013)* showed that the production capacity of growth-promoting products of *Bacillus* sp. isolated from sugarcane was higher than that of *Pseudomonas* sp. It is suggested that different *Pseudomonas* strains from different hosts vary in IAA-accumulation capacity. Moreover, all of these growth-promoting products were measured in vitro in this study and may be different in vivo because the specific functions of reNFB in living plants are also influenced by plant physiological and environmental pressures (*Moseman et al., 2009*).

Notably, the Shannon diversity index was negatively correlated with the productivity of all growth-promoting products, and all were statistically significant, with the exception of IAA in 144 h (Fig. 4). Considering that the number of reNFB isolated from each *A. adenophora* was similar, we speculated that even though the microbes had functional redundancy (*Miki, Yokokawa & Matsui, 2014*), the infection of functional microbes to host plants still had a certain threshold limit, by which the deficiency of microbial functions for host growth had to be compensated by improving their diversity or vice versa, resulting in a trade-off between quantity and diversity of functional microbes assembled within plants. There is some evidence for this concept in a previous study, where *Torres-Cortés et al. (2018)* suggested that nutrient availability during germination elicited changes in the composition of microbial communities by potentially selecting microbial groups with functional traits linked to copiotrophy. Thus, it represents a compelling field for further investigation of the interaction between reNFB and *A. adenophora* in vivo.

In the inoculation experiment, we verified the positive effect of reNFB on the seedlings of *A. adenophora* (Fig. 5). The isolate WS05 (*Rhizobium* sp.) revealed the best growth-promoting effect, while its IAA accumulation, phosphorus solubilization, and siderophore production was relatively low among the six inoculated isolates. The results indicated that the growth-promoting products of reNFB were not the crucial factors influencing seedling growth of *A. adenophora* in the conditions of this experiment. However, the diversified NFB may differ from one another in terms of the other functions they perform within an ecosystem, and the changes in habitat conditions may also lead to changes in the main mechanism of microbial effects. For example, *Moseman et al. (2009)* indicated that NFB showed various physiological and ecological functions in three wetland invasion systems. There is no doubt that the invasive process of plants will encounter many unknown environments. At this time, the multifunctional characteristics of reNFB are particularly important. Moreover, the positive effect of reNFB on the roots of *A. adenophora* seedlings was stronger than that on the shoots (Fig. 6). The reason for this finding may be that most endogenous bacteria have tissue specificity (*Lodewyckx et al., 2002*), which means that reNFB likely produce various growth products to function preferentially in the roots.

In this study, the positive effect of endophytic nitrogen-fixing bacteria on *A. adenophora* growth was proposed. However, native plants also commonly harbor endogenous nitrogen-fixing bacteria with positive effects (*Olivares et al., 1996*). Thus, whether the endogenous nitrogen-fixing bacteria enriched by invasive plants and native species are different, both in the species composition and in the growth-promoting of their hosts, must be further compared. A previous study on the competition of *A. adenophora* with native species in new habitats mainly involved its leaf litter allelopathy

(*Inderjit et al., 2011*). Therefore, it would be interesting to determine whether the secondary metabolites of *A. adenophora* or soil type in the invaded sites influence the population dynamics of rhizosphere and endophytic nitrogen-fixing bacteria, which in turn enhance the growth advantage of *A. adenophora*. In addition, for each sampling region, only three plant individuals were collected, and nearly 30 bacteria were isolated in this study. Because the species rarefaction curve did not reach a plateau (Fig. S1), it is expected that new OTUs will be discovered if more isolates are taken. Therefore, more strains should be isolated; particularly, new techniques such as high-throughput sequencing should be performed to indicate the reNFB diversity of *A. adenophora* in Yunnan Province. Physiological and morphological characteristics should also be taken into taxonomic classification of strains because solely using 16S rRNA gene sequences is not reliable below genus level. More importantly, it is worth determining the nitrogenase activity capacity as well as the nifH gene diversity for these reNFB to establish the direct relationship between nitrogen-fixing bacteria and *A. adenophora* invasion.

## CONCLUSIONS

In conclusion, there were a variety of endophytic NFB inhabiting the roots of *A. adenophora*, and the dominant genera ranked by number were *Pseudomonas*, *Rhizobium*, and *Duganella*. The number of reNFB was similar across different geographical regions, but the composition and diversity were markedly different. In addition to nitrogen fixation, the reNFB also distinctly accumulated IAA, dissolved phosphate and produced siderophores. Furthermore, plants with high reNFB diversity generally had low growth-promoting product productivity. The inoculation experiment verified that reNFB had a significant positive effect on the growth of *A. adenophora* seedlings, particularly on root growth. It is necessary to carry out further studies on the response of *A. adenophora* to reNFB under various environmental conditions.

## ACKNOWLEDGEMENTS

Authors thank Huan Yang, Yi-Shan Chen, Yi-Fang Miao, Tian Zeng, and Wen-Ti Zheng at Yunnan University, for partially sampling in wild and experiment performance.

### Funding

This work was supported by the National Natural Science Foundation of China (Nos. 30560033, 31770585). The funders had no role in study design, data collection and analysis, decision to publish, or preparation of the manuscript.

### Grant Disclosure

The following grant information was disclosed by the authors:
National Natural Science Foundation of China: 30560033 and 31770585.

### Competing Interests

The authors declare that they have no competing interests.

## Author Contributions

- Kai Fang conceived and designed the experiments, analyzed the data, contributed reagents/materials/analysis tools, prepared figures and/or tables, authored or reviewed drafts of the paper, approved the final draft.
- Zhu-Shou-Neng Bao conceived and designed the experiments, performed the experiments, analyzed the data, contributed reagents/materials/analysis tools, prepared figures and/or tables, approved the final draft.
- Lin Chen analyzed the data, prepared figures and/or tables, approved the final draft.
- Jie Zhou performed the experiments, approved the final draft.
- Zhi-Ping Yang performed the experiments, approved the final draft.
- Xing-Fan Dong performed the experiments, approved the final draft.
- Han-Bo Zhang conceived and designed the experiments, analyzed the data, contributed reagents/materials/analysis tools, prepared figures and/or tables, authored or reviewed drafts of the paper, approved the final draft, supported by foundation.

## DNA Deposition

The following information was supplied regarding the deposition of DNA sequences:
    Data is available at NCBI GenBank via accession numbers: MK249666–MK249688.

## Data Availability

    The raw measurements are available in the Supplemental File. The raw data shows all data used in the analysis of the figures and table.

## Supplemental Information

Supplemental information for this article can be found online at http://dx.doi.org/10.7717/peerj.7099#supplemental-information.

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
