# Peer review of "Growth-promoting characteristics of potential nitrogen-fixing bacteria in the root of an invasive plant Ageratina adenophora"

_PeerJ, doi:10.7717/peerj.7099_

## Round 0.1 · original submission · Major Revisions

The reviewers have highlighted several issues that must be addressed before this manuscript can be considered for publication.

One is the English editing which is required; this will take some additional but extensive work.

The issue of identification of microbes of interest; use of techniques other than 16S promoter for ID at the species level.

In addition, more detailed information and rationale is needed in the materials and methods to justify why you performed techniques or experimentation in a particular way, especially when it deviates from the industry standard.

Reviewer 1 ·

Basic reporting

no comment

Experimental design

l 151. Root sterilization is normally performed with intact roots. After sterilization, the roots are subjected to fragmentation. Root fragments may spill off internal contents and the sterilization process may reach the root interior.

l 154. Why did the authors used NA culture medium to check for surface sterilization, while using YMA to isolate reNFB? Please, clarify.

l 156-163. A common practice when isolating root bacteria is to proceed serial dilution from root extract. Normally, endophytic bacteria are isolated in 10^-5 to 10^-8 dilution. Why did not the authors proceed with serial dilution?

l 167-171. Was the cultivation of isolates made in nitrogen-free solid medium? Most of endophytic-nitrogen fixing bacteria are not able to growth (fixing nitrogen) in these conditions due to contact with high concentration of oxygen in the air. It includes taxonomic groups identified in this work. The normal procedure is to use semi-solid media. Please, clarify the procedure.

l 179. What were the reagents and concentrations in PCR mix reaction tube?

l 187-189. How exactly were the sequences filtered for chimeric bases? Why was variation in sequence length not allowed? "...cut out chimaeric bases to make each sequence 1351 pb long...". Does the 1351 represent the alignment length (in alignment positions), instead of sequence length (in bp)?
How many isolate sequences were generated? From OTUs in supplementary material (peerj-33237-Data.xlsx) it seems that 131 sequences were produced. However, in GenBank database only 23 sequences are available. Why? Although sequences may be very similar, they represent different isolated colonies. Also, clustering sequences at 97% identity means that differences exist within clusters that should be showed. Please, make all sequences available in GenBank database and provide all accession numbers in the manuscript.
How was the clustering analysis performed (e.g., software used)?
Versions and references for software and databases must be provided.

l 194. Taxonomic identification based on sequence similarity search is sensible to database content. What was the database selected for similarity search analysis (e.g., nr, 16S ribosomal RNA)? It seems that the taxonomic identification is based on best-blast hit. The authors must make it clear, specifying the database used and its version (or date). For example, strain CY25 was identified as Variovorax gossypii, but similarity search using megablast and "16S ribosomal RNA (Bacteria and Archaea)" show Variovorax guangxiensis as best hit. Strain WS26 was classified as Nocardioides albertanoniae, but its best hit is Nocardioides luteus. The authors should check for other differences. Taxonomic classification based solely on 16S rRNA gene sequences is not reliable below genus level. Other tools and databases may be used and the results showed in a comparative way.

l 250-251. "Tree cropland soils mixed with two humus and one vermiculite...". Do the values mean proportions of each substrate used? Please, clarify. Consider change the statement to (or equivalent)... "substrate was prepared including cropland soils, humus, and vermiculite, mixed in a proportion of 3:2:1".

Validity of the findings

In order to have the results clear and completely trustable, the authors need to clarify some methodological concerns and answer the specific questions, pointed out in this review.

Additional comments

l 134. The term "growth-promoting regulators" is better applied to refer only to phytohormones (it not include phosphate solubility and siderophore production).

l 293. Several isolates are close related to well known nitrogen-fixing bacteria within Alphaproteobacteria and Betaproteobacteria (e.g., Burkholderia and rhizobia). Although, those isolates were able to growth during 7 generations in nitrogen-free media, to confirm them as nitrogen fixer, more evidence should be provided, such as nifH amplification/sequencing and nitrogenase activity assay.

l 307-308. For each sampling region, three plants were collected and about 23 to 29 bacteria were isolated (only considering those with 16S rRNA gene sequence produced). Is this a representative sample in terms of diversity? If more isolates were taken, could new OTUs be discovered (e.g., rarefaction curve reach a plateau or not)? If not, Shannon index values may be biased.

Figure 1 (legend). "The bootstrap values were indicated at branch node, shown as MrBayes/Maximum Likelihood". Bootstrap analysis do not apply to Bayesian method.

Reviewer 2 ·

Basic reporting

The authors present findings on root endophytic nitrogen-fixing bacteria in an invasive weed Ageratina adenophora through isolation, characterisation and identification, followed by the assessment of growth promotion of Ageratina adenophora seedlings through soil inoculation of selected isolates. While the authors present a fairly comprehensive study, it is the opinion of this reviewer that further major revisions are required.
As a general comment, the manuscript requires English language editing. I have found numerous grammatical errors and confusing statements throughout. Some of these statements will be listed in the specific comments below. Therefore it is highly recommended that the authors seek English language corrections with the aid of a professional editor or native English speaker.
There is an excessive use of abbreviations in the manuscript that have not been explained at first mention. Please note that the readers of this journal are from various aspects of biology, medicine and life sciences and may not be aware of certain abbreviations, which the authors may consider common knowledge.
The authors should acknowledge the role of plant secondary metabolites influencing population dynamics of rhizosphere and endophytic nitrogen-fixing bacteria and their potential roles in plant-microbial communication. This information is lacking.
Specific comments
Ln 85-86 this sentence is confusing. Please reword for clarity
Ln 89 “mutualisms” suggest changing to mutualistic relationships
Ln 173 YMA should be YM as agar is not used in liquid culture
Ln 179 “MIX” please note the name of the kit used and mention manufacturer so that the experiment can be reproduced properly by someone following this method.
Ln 226 YMA, should this be YM as well?
Ln 258 “(day per night)” should read (day/night)
Ln333 “P=0.0000”, please check your P values
Please use the specific comments as guides to correct numerous other grammatical errors. The study does isolate and characterise a number of root endophytic nitrogen-fixing bacteria that also accumulate IAA, solubilise phosphorus and produce siderophores as measures of plant growth promotion. However, a direct impact of nitrogen fixation or plant growth promotion has not been deduced from these experiments.

Experimental design

The experiments presented in the manuscript fit within the aims and scope of the journal. The gaps in current knowledge have been identified by the authors and addressed in the presented experiments. The methods have been explained in sufficient detail to replicate with the exception of the use of unexplained abbreviations and the lack of references to reagents as mentioned above.
When characterising the microbial profiles from isolates in Ageratina adenophora, it is also important to consider the endophytic microorganisms present in key native plants. As one of the claims of the study is that there is potentially recruitment of reNFB by Ageratina adenophora, a direct comparison should be made with a native plant/ plants, or as a minimum discussed in the manuscript as a limitation of the study
Specific comments
Ln 140 Authors should identify the specific locations the plant samples were collected and explain the soil properties including soil type, pH and conductivity as the soil type often has a major impact on plant growth characteristics and microbial populations

Validity of the findings

In general the findings have been statistically analysed satisfactorily to answer the research question. Plant growth promotion by the 6 isolates as tested in Ageratina adenophora seedlings was presented well and supports the hypothesis that the recruitment of reNFB may enhance plant growth promotion in this plant and potentially aiding in its invasiveness.

Specific comment: Figure 3- Comparison of study sites on plant growth promotion. Were there statistical differences between the sites? If there were no differences, please indicate as not significant.

---

## Round 0.2 · Minor Revisions

Please pay strong attention to the comments of reviewer 1. You have not yet presented strong rationale for WHY you chose to use the methodology you did to make the assumption that you were isolating and characterizing N fixing bacteria in this study. Without addressing this issue, it may be difficult to publish the work according to reviewer 1.

Reviewer 1 ·

Basic reporting

no comment

Experimental design

Although the methodology is better presented and clear in this version of the manuscript, the methods used for endophytic bacteria isolation was not adequate and do not guarantee the isolation of nitrogen-fixing bacteria. No further evidence for nitrogen fixation was provided. However, the authors stands the work on "root endophytic nitrogen-fixing bacteria". Further, there was not a quantification of the endophytic bacteria.
Although phylogeny is only used as a way to cluster the isolates, once the authors chose to use it, it must be performed in a correct way. But the tree showed in the manuscript suffer from some conceptual/methodological problems.
Please, see details in the attached file.

Validity of the findings

Please, see observations in "Experimental design" above, that interfere in the validity of the findings and other details discussed in the attached file.

Additional comments

Please, see the reply for all authors responses (first review) in the attached file.

Annotated reviews are not available for download in order to protect the identity of reviewers who chose to remain anonymous.

Reviewer 2 ·

Basic reporting

The clarity of the information contained within the manuscript has been enhanced through professional English editing and all queries directed to the authors have been addressed in the manuscript

Experimental design

Queries on the experimental design especially with respect to 16rRNA analysis and media selection have been addressed by the authors

Validity of the findings

Discussion of the limitations of the manuscript have been adequately addressed by the authors and errors in statistical analysis corrected.

Additional comments

Thank you for providing edits to the manuscript as requested. The manuscript has improved in quality when compared to the initial submission. I believe the comments I have provided have been addressed adequately.

---

## Round 0.3 · accepted · Accept

Thanks for addressing the concerns of all reviewers in this version.